# Interlayer exciton mediated second harmonic generation in bilayer MoS₂

Shivangi Shree[1,2], Delphine Lagarde[1], Laurent Lombez [1], Cedric Robert[1], Andrea Balocchi[1], Kenji Watanabe [3], Takashi Taniguchi [4], Xavier Marie [1], Iann C. Gerber [1], Mikhail M. Glazov [5✉], Leonid E. Golub [5], Bernhard Urbaszek [1✉] & Ioannis Paradisanos [1✉]

Second-harmonic generation (SHG) is a non-linear optical process, where two photons coherently combine into one photon of twice their energy. Efficient SHG occurs for crystals with broken inversion symmetry, such as transition metal dichalcogenide monolayers. Here we show tuning of non-linear optical processes in an inversion symmetric crystal. This tunability is based on the unique properties of bilayer MoS₂, that shows strong optical oscillator strength for the intra- but also interlayer exciton resonances. As we tune the SHG signal onto these resonances by varying the laser energy, the SHG amplitude is enhanced by several orders of magnitude. In the resonant case the bilayer SHG signal reaches amplitudes comparable to the off-resonant signal from a monolayer. In applied electric fields the interlayer exciton energies can be tuned due to their in-built electric dipole via the Stark effect. As a result the interlayer exciton degeneracy is lifted and the bilayer SHG response is further enhanced by an additional two orders of magnitude, well reproduced by our model calculations. Since interlayer exciton transitions are highly tunable also by choosing twist angle and material combination our results open up new approaches for designing the SHG response of layered materials.

[1] Université de Toulouse, INSA-CNRS-UPS, LPCNO, 135 Avenue Rangueil, 31077 Toulouse, France. [2] Department of Physics, University of Washington, Seattle, WA, USA. [3] Research Center for Functional Materials, National Institute for Materials Science, 1-1 Namiki, Tsukuba 305-0044, Japan. [4] International Center for Materials Nanoarchitectonics, National Institute for Materials Science, 1-1 Namiki, Tsukuba 305-0044, Japan. [5] Ioffe Institute, 194021 St. Petersburg, Russia. ✉email: glazov@coherent.ioffe.ru; urbaszek@insa-toulouse.fr; paradeis@insa-toulouse.fr

Nonlinear optical spectroscopy of atomically thin crystals gives crucial insights into their physical properties and investigates potential applications[1,2]. Nonlinear optics is based on the interaction of photons within the crystal. This can give rise, among other processes, to second-harmonic generation (SHG) where two photons of identical energy combine into one photon of twice this energy[3,4]. Currently existing applications in photonics and laser physics make use of this process in bulk crystals for example using potassium dihydrogen phosphate (KDP) or lithium niobate (LiNbO$_3$)[5].

SHG is vital both as a spectroscopic tool[6–8] and also for applications of a large class of materials such as II-VI and III-V semiconductors[9–12], nanotubes[13], magnetic- and nonmagnetic layered materials[14–20], perovskites and antiferromagnetic oxides[21,22].

The occurrence of SHG in a crystal is directly linked to its symmetry, where crystals with broken inversion symmetry can exhibit SHG and crystals which are inversion symmetric can typically not[5]. This dependence on symmetry makes nonlinear optics in atomically thin crystals such as graphene a very rich field of research[23–29]. Although pristine graphene is an inversion-symmetric crystal, the crystal symmetry or symmetry of the electronic states is potentially very sensitive to charges or single molecules in the layer vicinity[30], an imbalance in population of valley states[31], or electric currents[32]. However, graphene is a gapless material, so light-matter interaction cannot be controlled to the same extent as in atom-thin semiconductors based on transition metal dichalcogenides (TMD).

TMD monolayers such as MoS$_2$ and WSe$_2$ have broken inversion symmetry and show intrinsically strong SHG signals[1,33,34]. Excitons are strongly bound in these materials[35] and govern optical processes also at room temperature, contrary to the model semiconductor GaAs[10]. This is crucial for nonlinear optics as excitons resonantly enhance light-matter interaction by several orders of magnitude[35]. As a result, in monolayers the excitonic contribution to SHG can be orders of magnitude higher than the intrinsic contribution from the crystal which exists also off-resonance[17,18,36–38]. An intrinsic challenge for exploiting SHG signals from nanostructures is the overall small signal, despite the giant response of a monolayer per unit thickness[34,39,40]. Hence, in the interest of both fundamental physics and applications, we need to uncover the exact origins of SHG in layered semiconducting materials to enable further tuning and amplification.

In this work we demonstrate strong, tunable SHG based on exciton resonances in nominally inversion-symmetric MoS$_2$ bilayers with 2H stacking: Varying the laser excitation energy allows us to address resonantly not only intralayer and but also interlayer exciton states with large oscillator strength[41], which makes MoS$_2$ bilayers particularly interesting. At these specific energies we generate SHG signals in MoS$_2$ bilayers which are (i) comparable in amplitude to off-resonance SHG monolayer signals and (ii) several orders of magnitude larger than reported in the literature for TMD bilayers with 2H stacking[33,38,42,43]. The prominent role of interlayer excitons in MoS$_2$ bilayers allows for an efficient SHG tuning mechanism: in applied electric fields, we lift the degeneracy of the interlayer exciton states through the Stark effect. This results in an additional enhancement of the SHG signal at the interlayer exciton resonance by two orders of magnitude, surpassing the response of an ungated monolayer at this energy. We uncover the excitonic origin of the highly tunable SHG response through the linear polarization dependence of the SHG signal with respect to the armchair direction of the sample. We compare bilayer results with measurements in monolayers. We uncover variable SHG enhancement in trilayers located in the same van der Waals stacks. We present a model analysis of the emergence of the strong SHG in the bilayer, taking into account

the impact of electric fields perpendicular to the bilayer as well as different sources of asymmetry.

## Results

**SHG in MoS$_2$ bilayers on SiO$_2$/Si.** Very recently, details of the rich excitonic resonances in bilayer MoS$_2$ have been uncovered[41]. We aim to distinguish the possible SHG generation that comes from a weak, intrinsic, crystal-related contribution from the possible excitonic contribution by using a tunable laser source. This allows comparing off-resonance with on-resonance experiments.

For the experimental results shown in Fig. 1, we investigate a 2H MoS$_2$ bilayer deposited on 90 nm of SiO$_2$ on top of a Si substrate, a very typical sample configuration (see Mtethods and Supplementary Information for details in the fabrication process). The bilayer sample (as well as the trilayer sample below) has been directly exfoliated from a 2H bulk crystal, i.e., without manual stacking individual layers. We ensure therefore 60° twist angle (and not 0° as for 3R stacking) between adjacent layers, as confirmed by measuring the energy separation of the A–B excitons and observing the interlayer exciton[44]. We use a pulsed Ti:Sapphire laser source coupled to an optical parametric oscillator (OPO, using 5 mW average power and 1 ps pulse duration of 80 MHz repetition rate correspond to ≃60 W peak power) and we scan twice the laser energy (i.e. $2 \times E_L$) over an energy range that corresponds to the main optical transitions in the MoS$_2$ bilayers, namely the interlayer and intralayer exciton resonances[45–48].

Experiments are performed at a temperature of $T = 4$ K in vacuum in a confocal microscope (excitation/detection spot diameter of the order of the wavelength), see supplement and[7] for details. We plot in Fig. 1b a series of SHG spectra from the bilayer for a number of selected laser energies between $E_L = 0.9$ and 1.13 eV in steps of ≈6 meV. Each SHG spectrum is a single peak that shifts in energy as we vary the excitation laser energy $E_L$. The spectral width of the SHG signal is limited by the laser pulse duration (ps).

We start with the off-resonance case: we do not detect any SHG signal within our typical integration time (2 minutes and excitation power of a few mW) when twice the laser energy ($2 \times E_L$) is tuned below the intralayer A energy (i.e. the lowest lying direct optical transition with large oscillator strength). But surprisingly for a crystal with inversion center, as we increase the laser energy, we see clear resonances in the SHG response, i.e. a strong variation in SHG intensity as a function of laser energy $E_L$ in Fig. 1b. In Fig. 1c, we plot the differential white light reflectivity spectrum of this sample on the same energy scale.

Comparing SHG spectra and white light reflectivity, we deduce that the maxima in the SHG intensity occur when $2 \times E_L$ is in resonance with the intralayer (A and B) and interlayer (IE) exciton energies[44].

The surprisingly strong SHG signal could suggest a symmetry breaking in our nonencapsulated sample. A possible reason can be a drastically different dielectric environment experienced by top and bottom MoS$_2$ layers from Fig. 1b.

**SHG in MoS$_2$ bilayers encapsulated in hBN.** To verify the impact of the dielectric environment, we have fabricated MoS$_2$ bilayer samples encapsulated in high quality hexagonal BN[49]. This more symmetric dielectric environment[50–52] eliminates potential sources of symmetry breaking. In addition, we also exclude the possibility of adsorbates or molecules at the sample surface in encapsulated samples which might impact the measurements as in surface SHG studies[53] and suggested in graphene[24,30]. In Fig. 1e we plot the SHG spectra as a function of twice the laser energy $2 \times E_L$ using a smaller

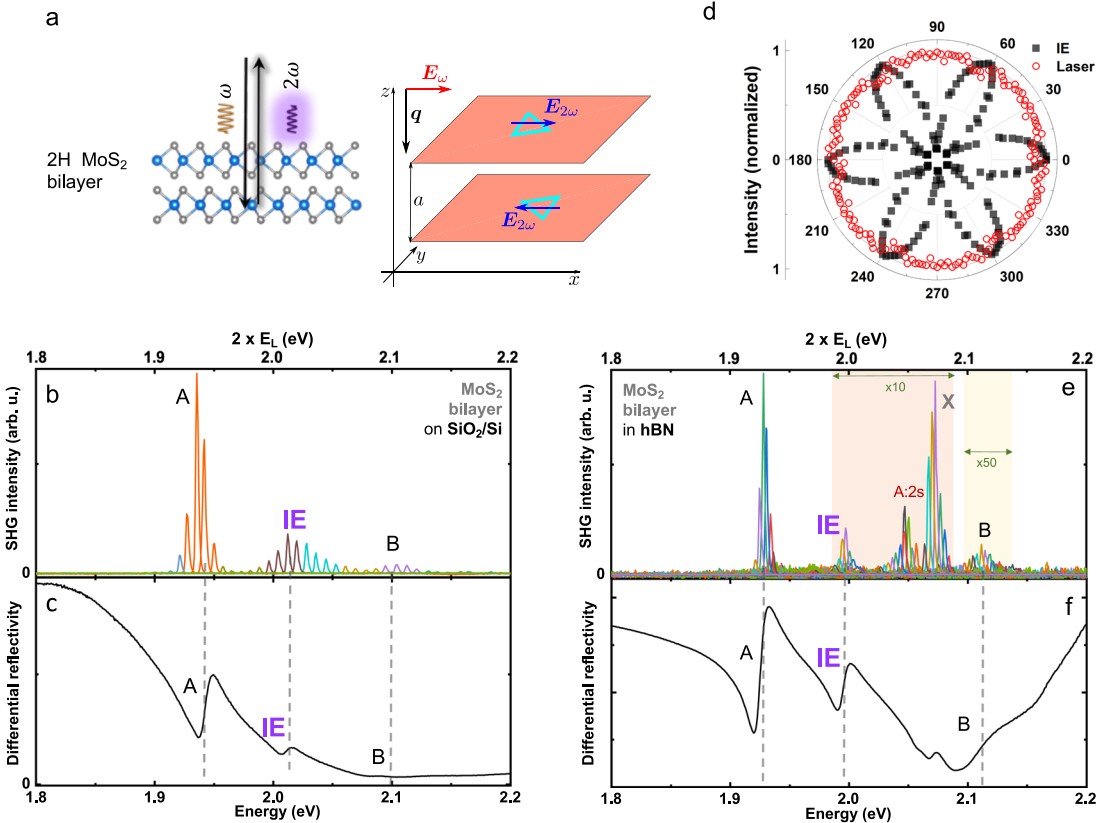

**Fig. 1 Second-harmonic generation (SHG) spectroscopy in bilayer MoS₂ with 2H stacking in different dielectric environments. a** Schematics of SHG in MoS₂ bilayers with 2H stacking. **b** SHG spectra plotted as a function of twice the laser energy $2E_L$. Each spectrum is a separate measurement and we plot all the SHG spectra for the different laser energies as in ref. [36]. SHG experiments are performed on nonencapsulated 2H MoS₂ bilayers (laser power 4 mW) and hBN-encapsulated samples (10 mW), see (**e**). The main excitonic transitions are marked: intralayer A, B and interlayer (IE). **c** Differential reflectivity spectrum of the same sample as a function of photon energy. **d** Polarization-resolved SHG intensity[33] for $2 \times E_L = 2.01$ eV in resonance with the interlayer exciton IE (black squares) energy of 2H MoS₂ bilayers. For comparison we show the laser reflection (red circles) which remains constant for each angle. (**e**, **f**) same as (**b**, **c**), but for bilayer samples encapsulated in hBN. Note that in (**e**) we have multiplied the SHG signal in the orange shaded region by 10, and in the yellow shaded region by 50 to discern all transitions on a linear scale. In addition to previously identified exciton transitions a new transition at 2.05 eV emerges in SHG spectroscopy, that we tentatively attribute to the A:2s state of the intralayer excitons. High field magneto-optics is needed to confirm this assignment[71] through measuring the diamagnetic shift.

step size of ≈3 meV between two adjacent SHG spectra. The SHG signal coming from hBN is several orders of magnitude weaker than from the MoS₂ bilayer, see[33] and our data in the supplement. Our experiments in Fig. 1e show that also hBN-encapsulated MoS₂ bilayers in a far more symmetric dielectric environment show SHG at the exciton resonances. Comparing the signal strength for the A-exciton resonance, we record an SHG signal about an order of magnitude weaker for the hBN-encapsulated sample (Fig. 1e) as compared to the nonencapsulated sample on SiO₂ (Fig. 1b). This indicates that indeed a sizeable contribution to the SHG in bilayers results from a very different dielectric environment for the top layer as compared to the bottom layer. The resonances are spectrally narrower in the encapsulated samples, see Fig. 1e, consistent with TMD exciton transitions in high quality hBN-encapsulated layers[54]. This spectral narrowing in combination with cavity effects in encapsulated samples will modulate the wavelength dependence of SHG intensity. Comparison with white light reflectivity data in Fig. 1f allows attributing the different interlayer and intralayer transitions. The slight overall shift in energy of the exciton transitions for encapsulated versus nonencapsulated samples comes mainly from renormalization of all the Coulomb energies as the effective dielectric constant is different for the two bilayer samples.

Our surprising finding on interlayer exciton-mediated SHG merits further investigation. In Fig. 1d, we plot the polarization dependence of the SHG response at the interlayer exciton resonance. We excite with linearly polarized light and the strength of the SHG signal collected in the same polarization depends on how the crystallographic axes are aligned with respect to the laser polarization. We clearly observe a 6-fold rotational symmetry expected for the space group of a 2H bilayer[33], see discussion below. This polarization dependence is a strong indication that the SHG signal is due to intrinsic effects linked to crystal and exciton symmetry. We have performed measurements at additional exciton resonances that give the same polarization dependence.

**Comparison of SHG in hBN-encapsulated mono-, bi-, and trilayers.** We provide in Fig. 2a–c results for hBN-encapsulated mono-, bi-, and trilayers, for the same sample used in ref. [41]. Our aim is to compare the surprising SHG amplitude for bilayers with the monolayer and trilayers, for which crystal inversion symmetry is broken and as a consequence strong SHG is expected[33]. Strikingly, for all the experiments on the mono-, bi-, and trilayers, we see that the SHG signal is orders of magnitudes enhanced when twice the laser energy $2 \times E_L$ is in resonance with an excitonic transition, as compared with a nonresonant situation.

Previous reports on SHG in MoS₂ bilayers did not focus on exciton resonances[1,33,38,42,43,55] and hence signals from monolayers were 3 orders of magnitude higher than for bilayers in the

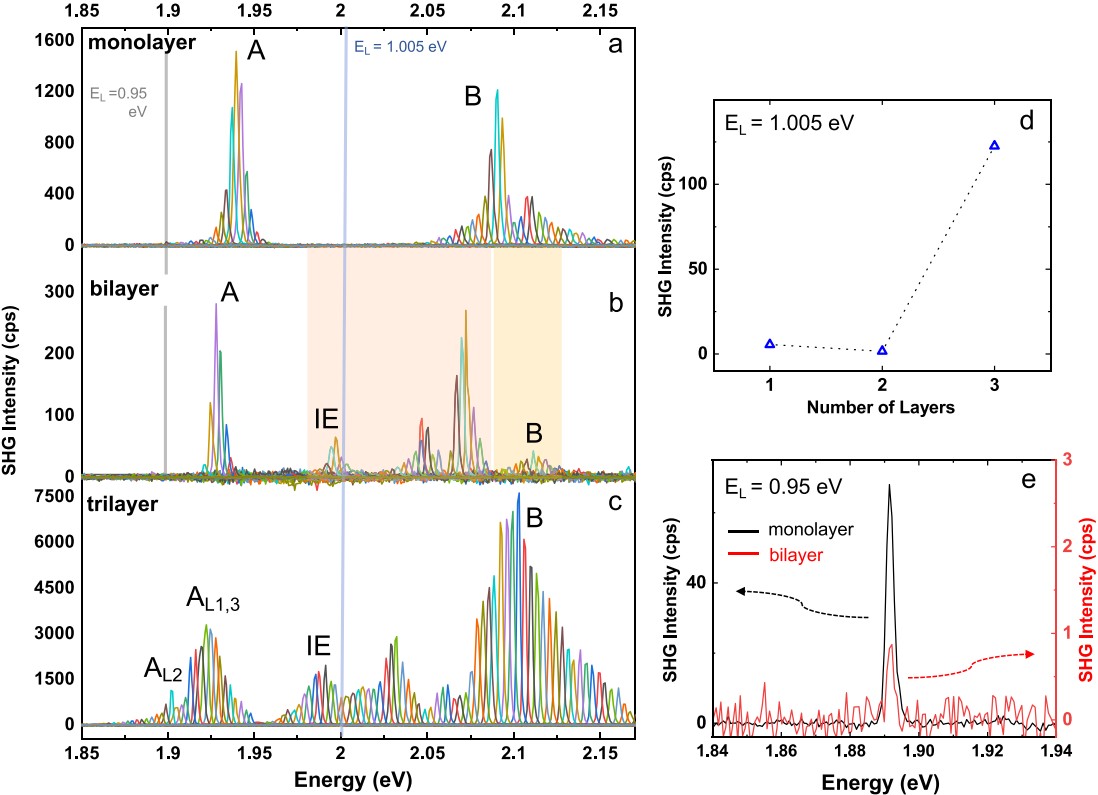

**Fig. 2 Comparing SHG spectroscopy for MoS₂ mono-, bi-, and trilayers.** Data from the same staircase flake of MoS$_2$ mono-, bi-, and trilayer encapsulated in hBN. **a** SHG spectra plotted as a function of twice the laser excitation energy $2E_L$ for the monolayer. The main exciton transitions are marked, compare with Fig. 1e,f for the bilayer and the supplement for mono-and trilayers. **b** SHG spectra for bilayer, where intensity in orange shaded region is ×10, yellow shaded region ×50. **c** SHG spectra for trilayer. The counts in panels a, b and c can be used as a direct comparison for the SHG amplitude from mono-, bi-, and trilayer recorded under identical conditions. **d** SHG intensity compared for the same power density used on the same encapsulated sample for a laser energy of $E_L = 1.005$ eV i.e. in resonance with IE in the bilayer, see panels (a-c), bright blue vertical line. The monolayer signal is only a factor of 3 stronger than the bilayer. **e** For a laser energy of $E_L = 0.95$ eV (indicated by vertical bright gray line in **a** and **b**) we plot the emission spectra in the spectral vicinity of $2E_L$ for the monolayer (black) and for the bilayer (red) using the left and right y-axis, respectively.

off-resonant case. From our measurements in panels Fig. 2a, b we deduce that $E_L = 0.95$ eV (i.e. SHG energy of 1.9 eV) is below the A-intralayer exciton resonance for monolayers and bilayers. We directly compare the measured SHG signal at this laser energy for monolayers and bilayers, see peaks in Fig. 2e, and find that the monolayer signal is indeed 2 orders of magnitude larger than the bilayer signal. The situation is drastically different as we change laser energy: At $E_L = 1.005$ eV twice the laser energy $2 \times E_L$ is in resonance with the interlayer IE transition of the bilayer resulting in a strong amplification of the SHG signal while being nonresonant for a monolayer, see comparison in Fig. 2d. These different situations with respect to the exciton resonance result in a comparable overall SHG amplitude for mono- and bilayers.

Although SHG signals in trilayers higher than in bilayers have been reported at fixed laser energy, very little is known of the energy dependence of the nonlinear susceptibility of the trilayer. The trilayer crystal has no inversion center, as the monolayer, and sizeable nonresonant SHG can be expected. In addition, there exist several intralayer and interlayer exciton resonances[41] that can potentially enhance SHG. For the measurements in Fig. 2c, we find an SHG signal with strong amplitude variations over the investigated energy range 1.87 and 2.15 eV. We identify the two intralayer transitions for excitons in the central layer ($A_{L2}$) and in the outer layers ($A_{L1}$ and $A_{L3}$) respectively, between 1.9 and 1.92 eV. Local maxima around 1.98 eV can be attributed to interlayer excitons, with B-exciton contributions around 2.1 eV.

From our measurements, we conclude that the trilayer combines the high intrinsic SHG response of the monolayer with the tunability (see below) of the interlayer exciton-mediated SHG. The exciton transitions for mono- and trilayers are identified and labeled in more detail by comparing with white light reflectivity[41] shown in the supplement.

The A-exciton resonance in trilayers for the middle layer $L_2$ is redshifted compared to the A-exciton resonance for the outer layers $L_1$ and $L_3$. Because of this shift, these contributions do not cancel each other as could be expected from symmetry if they would be at the same energy. The A-exciton of the monolayer gives about 1500 counts (Fig. 2a), the A-exciton of $L_2$ gives also about 1500 counts and at the energy of A-exciton for $L_1$ and $L_3$ we get about 3000 counts (Fig. 2c). If for the trilayer the A-excitons in all 3 layers would be degenerate, we would expect to measure roughly only $3000 - 1500 = 1500$ overall counts for the A-excitons. But in an MoS$_2$ trilayer this degeneracy among the A-excitons is lifted and we record in total 4500 counts in Fig. 2c.

In addition to measurements at $T = 4$ K, we have carried out SHG spectroscopy for mono-, bi-, and trilayer MoS$_2$ also at room temperature, see supplement. We observe the typical broadening of the excitonic transitions and the bandgap shift to lower energy[7]. It has been reported that the excitonic enhancement of the SHG signal also applies to room temperature experiments on monolayers[56,57]. Here we show that this is also true for MoS$_2$ bilayers and trilayers when twice the laser energy is tuned into

resonance with interlayer and intralayer excitons, underlining the importance of exciton-mediated SHG. This is of practical relevance as changing the laser energy in order to enhance the SHG signal is simpler in some experiments than changing the sample or device temperature.

In agreement with earlier estimations[58,59], we find a power of the SHG emitted by the monolayers of the order of a few to 10 fW for 3 mW laser excitation power. For the bilayer, the maximum measured SHG power is of the order of 1 fW for 5 mW excitation. For the trilayer we find similar power as for the monolayer. We discuss the sheet polarizability and the conversion efficiency in more detail in the Supplementary Information.

**Interlayer exciton-mediated SHG tuning**. We now show that by applying an electric field normal to the bilayer, we can further increase the SHG signal specifically at the interlayer exciton resonance. The interlayer exciton has an in-built, static electric dipole. In the absence of electric fields, there are two degenerate interlayer exciton configurations, with a hole delocalized over both layers and an electron in the top or bottom layer, respectively, as shown in Fig. 3d. This strong absorption feature is highly tunable in energy through the Stark effect[41,48,60]. In Fig. 3, we show that the SHG signal is also tunable in amplitude and also spectrally when an external electric field is applied to the bilayer. In previous works, the effect of the electric field application on SHG response of 2H bilayers[42,43] has been addressed only in terms of the impact of doping and crystal symmetry breaking, while interlayer exciton tuning as in our resonant SHG experiment has not been observed.

In Fig. 3a, we compare the SHG signal of a bilayer with and without an applied electric field. For this gated sample, the IE SHG signal reaches 5 counts/s at $F_z = 0$. As we apply a field of $F_z = 0.17$ MV/cm this signal increases by a factor of 25 to about 125 counts/s at the IE resonance maximum. To study this tuning

in more detail, we plot in Fig. 3b the SHG amplitude as a function of the applied electric field for six separate experiments. We see a quadratic increase of the IE SHG signal as a function of applied electric field $F_z$. This quadratic increase is due to mixing with intralayer excitons, as we show below.

In contrast, for the intralayer excitons, we do not record any significant increase of the SHG amplitude as $F_z$ is increased. At $F_z = 0.17$ MV/cm the SHG resonance shows a slight red-shift (Fig. 3a), possibly indicating charging effect, i.e., a shift toward trion transitions. Strong charging effects as a possible source of symmetry breaking in bilayer SHG response are discussed for WSe$_2$[43]. In our experiments, the target is to apply a static electric field and not to generate charging effects. As we trace the evolution of the neutral interlayer exciton as a function of the applied field, we rely on the Stark effect and not charging effects for the increase in SHG amplitude.

We now turn our attention to the spectral range over which the SHG enhancement occurs. In a 2H homobilayer at $F_z = 0$ two degenerate IE configurations exist: with a hole delocalized over both layers bound to an electron either in the bottom (IE$_1$) or top (IE$_2$) layer, see sketch in Fig. 3d. Application of a nonzero $F_z$ lifts the degeneracy of IE$_1$ and IE$_2$ as their static, permanent dipoles point in opposite directions. This leads to a Stark shift to lower and higher energy, respectively[41]. This lifting of the degeneracy can be seen in Fig. 3c in white light reflectivity experiments. We extract a splitting between IE$_1$ and IE$_2$ of roughly 10 meV for $F_z = 0.13$ MV/cm. In Fig. 3e, we plot SHG spectroscopy results at the IE resonance for $F_z = 0$, 0.07 and 0.13 MV/cm, respectively. In addition to the increase in SHG amplitude with $F_z$, we therefore observe that the spectral range for which amplification is observed is widened. This broadening is the consequence of the energy splitting between IE$_1$ and IE$_2$ transitions in an applied electric $F_z$. The broadening can also be seen directly in Fig. 3a. Tuning both amplitude and spectral range of the SHG response of

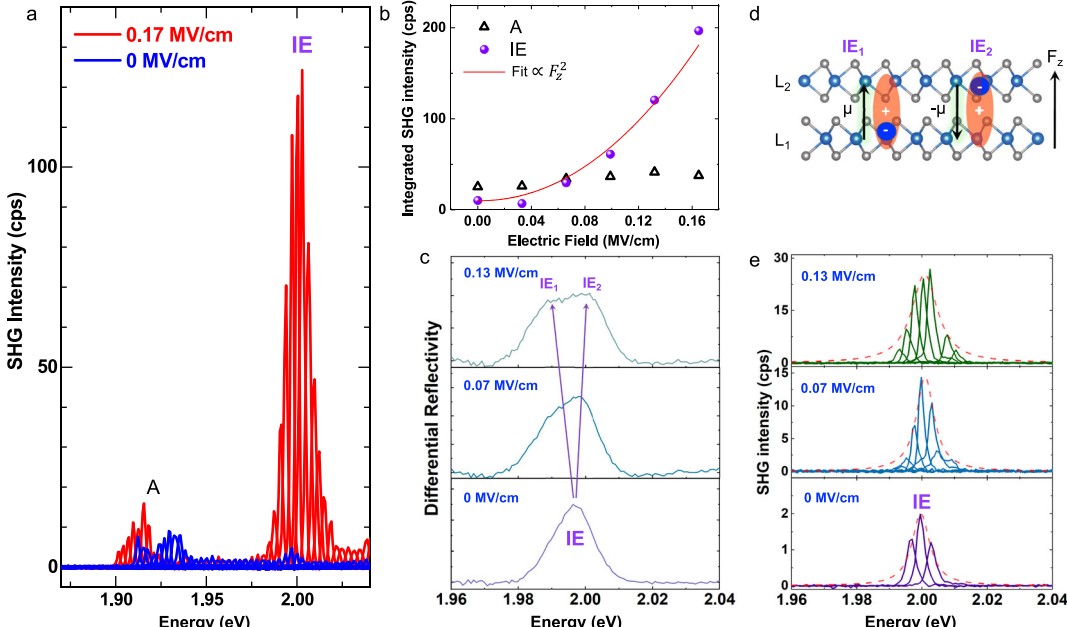

**Fig. 3 Tuning the SHG signal of a 2H MoS$_2$ bilayer in applied electric fields—Stark splitting.** Sample with contacts to apply vertical electric field $F_z$. **a** SHG spectra from a gated bilayer, covering intralayer A and interlayer IE exciton resonances at $F_z = 0$ MV/cm (blue spectra) and $F_z = 0.17$ MV/cm (red spectra) as a function of twice the laser energy $2E_L$. **b** Comparison of SHG signal for interlayer exciton IE (solid circles) versus intralayer exciton A (open triangle) as a function of applied electric field $F_z$. The SHG amplitude of IE is fitted with a quadratic function $\propto F_z^2$ (red line). **c** White light reflectivity for three different electric field values $F_z = 0$, 0.07, 0.13 MV/cm. Here we plot the first derivative of the reflectivity spectra for a better estimation of the IE energies. **d** Schematic of a bilayer with 2 distinct interlayer configurations, that have opposite permanent electric dipoles $\pm\mu$. **e** SHG spectroscopy for $F_z = 0$, 0.07, 0.13 MV/cm tuning twice the laser energy $2 \times E_L$ across the interlayer exciton resonance IE.

the bilayer is a direct consequence of exploiting the properties of interlayer excitons applicable to many van der Waals bilayer systems[61–66] and will be further discussed below.

## Discussion

We first focus on the model considerations for SHG mediated by the interlayer excitons in bilayer MoS$_2$. Here the SHG signal is very sensitive to the electric field, being one of the most striking experimental observations. Further we briefly address other contributions to the SHG. The detailed theory is presented in the supplement.

A TMD monolayer is described by the noncentrosymmetric point group $D_{3h}$ which lacks an inversion center and allows for the SHG. An ideal homobilayer in absence of external fields has an inversion symmetry and is described by the $D_{3d}$ point group where the SHG is forbidden. Qualitatively, this is because the constituent monolayers in the bilayer are rotated by an angle of $\pi$ with respect to each other in 2H stacking[67,68], and the electric field at a double frequency generated in one monolayer is compensated by the contribution of the other, Fig. 1a.

An electric field $\boldsymbol{F}$ perpendicular to the monolayers breaks the symmetry and enables the SHG described by a phenomenological relation

$$P_{2\omega,x} = \chi(E_{\omega,x}^2 - E_{\omega,y}^2), \quad P_{2\omega,y} = -2\chi E_{\omega,x}E_{\omega,y}, \quad (1)$$

where $E_\omega$ is the incident field and $P_{2\omega}$ is the polarization induced at a double frequency, $x$ and $y$ are the main in-plane axes of the structure. Since the bilayer is centrosymmetric, the nonlinear susceptibility $\chi$ in Eq. (1) changes sign at the space inversion, particularly, $\chi(-F_z) = -\chi(F_z)$. This relation reflects the three-fold symmetry of the bilayer where the quadratic components of the vector component $E_x^2 - E_y^2$, $-2E_xE_y$ transform as vector components $P_x$, $P_y$, see ref. [37] for details. Polarization dependence of the SHG predicted by Eq. (1) is observed experimentally, Fig. 1(d), showing sixfold symmetry of the crystal structure.

Intralayer excitons show negligible Stark shifts in applied fields[41], as their static dipole moment is negligible, as for excitons in separate monolayers[69,70]. Therefore, due to their strong in-built electric dipole, the main effect of the field $F_z$ is on the indirect excitons IE$_1$ and IE$_2$ whose degeneracy is lifted yielding $E_{IE_i} = E_{IE} \mp F_z\mu$, where $\pm\mu$ is the static dipole moment of the corresponding IE, Fig. 3(d). As a result of the energy splitting (Fig. 3c), the SHG contributions from IE$_1$ and IE$_2$ do not cancel, resulting in a strong SHG signal as the energy separation between IE$_1$ and IE$_2$ increases (Fig. 3a). The second-order susceptibility in the vicinity of the IE resonances in the linear-in-$F_z$ regime can be calculated taking into account two-photon excitation and coherent one-photon emission from the exciton states[37] as (see supplement for details):

$$\chi_{IE} = -2\mu F_z \frac{|T|^2}{\Delta^2} \frac{C_2|\Phi_{B:1s}(0)|^2}{\left(2\hbar\omega - E_{IE} + \frac{|T|^2}{\Delta} + i\Gamma_{IE}\right)^2}. \quad (2)$$

Here $C_2$ is the combination of the interband momentum matrix element, electron charge, free-electron mass and the band gap, $\Phi_{B:1s}(0)$ is the envelope function of the B:1s exciton at the coinciding electron and hole coordinates, $T$ is the hole tunneling matrix element between the layers, $\Delta = E_B - E_{IE}$ is the energy splitting between the IE and B-excitons (which are mixed due to the hole tunneling[45,47]), and $\Gamma_{IE}$ is the damping of the IE resonance. As demonstrated in previous works[41,60], the IE exciton optical activity is mainly due to the mixing with B-excitons, thus, in Eq. (2) the ratio $|T/\Delta|^2$ accounts for the IE-B exciton mixing in the two-photon excitation and single-photon emission channel. Equation (2) holds provided $|\mu F_z| \ll \Gamma_{IE}$, otherwise two peaks in

the SHG spectra are expected to split by $2|\mu F_z|$ (see supplement for details).

Two important conclusions can be drawn from Eqs. (1) and (2): First, $P_{2\omega}$ is maximized when twice the laser energy is resonant with an indirect exciton state as $2 \times E_L = 2\hbar\omega = E_{IE}$. Second, the intensity of the SHG scales as $F_z^2$. Experimental SHG intensities in Fig. 3a, e as well as in Figs. 1b, e and 2b are strongly enhanced at the IE resonance. In the measurements in Fig. 3b the SHG amplitude grows quadratically as a function of the applied $F_z$. So both experimental observations strongly support our analysis.

Importantly, the experiment demonstrates also the SHG on other excitonic species in bilayers, including weakly field-dependent effect on intralayer A-excitons, Fig. 3(a, b), and also the SHG at zero field on intralayer A:2s and B-excitons. A weaker effect of the electric field $\boldsymbol{F}$ on the A-excitons can be readily understood taking into account the fact that the A-excitons are mixed, e.g., with the interlayer B-excitons via hole tunneling. Corresponding energy distance $\Delta' = E_{IE(B)} - E_A \gg \Delta$ resulting in a weaker susceptibility to $F_z$.

We now discuss the enhancement of the SHG at the intralayer exciton resonances in the nominally centrosymmetric situation when $F_z = 0$. Possible origin of SHG in this case could be the quadrupolar or magneto-dipolar SHG, where akin to graphene case[23,27], the effect is related to the light wavevector $q_z$ at the normal incidence. The symmetry analysis again yields Eq. (1) with $\chi \propto q_z$. Particularly, a (tiny) phase difference of $\phi \sim q_z a$ where $a$ is the interlayer distance in the bilayer yields an imbalance of the contributions of individual monolayers and results in the resonant contributions at A- and B-excitons similar to calculated in[37] for monolayers, but being by a factor of $\phi \sim 10^{-2}...10^{-3}$ smaller (see supplement). Our estimates show, however, that this effect is too small to account on its own for the surprising experimental findings which demonstrate that the resonant susceptibilities of the bilayer are roughly one order of magnitude smaller than those of the monolayer.

Thus, we arrive at the conclusion that, despite relatively symmetric environment of our bilayer sample, the structure lacks an inversion center. Possible options are (i) small built-in electric fields and (ii) inequivalence of the intralayer excitons in monolayers forming a bilayer. Option (i) results in the replacement of $F_z$ by $F_z + F_0$ where $F_0$ is the normal component of the build-in field. In this case at $F_z = -F_0$ the effectively symmetric situation can be realized. While the measured dependence of the SHG intensity for IE, Fig. 3, does not contradict this scenario, note the minimum at $F_z \approx -0.035$ MV/cm, the SHG intensity measured at the A-exciton resonance does not significantly drop in this electric field range. Alternatively, we may suppose that the build-in field is inhomogeneous in the sample plane within our detection spot and the contributions of the A-exciton and IE states come from slightly different nanoscopic regions.

Option (ii) implies that the energies of A- and B-excitons, their nonradiative broadening, or their oscillator strengths are inequivalent in the two monolayers that form the bilayer. This may be due to the dielectric disorder, which, although suppressed, can still be present in state-of-the-art, hBN-encapsulated samples[51]. In this situation, the IE is mainly activated by the electric field, while intralayer excitons are less sensitive to $F_z$ (see supplement). Inhomogeneous broadening effects due to disorder, impurities, etc. are inevitable and inhomogeneities on the order of one micrometer can occur even in hBN-encapsulated samples[51,52]. For the in-plane disorder the crucial parameter is the ratio between the size of the inhomogeneity, $l$, and the laser spot diameter, $\delta$. For $l \ll \delta$, inhomogeneous broadening effects will effectively cancel out and the structure will effectively maintain the space inversion. However, for $l \gtrsim \delta$, the inhomogeneities will lift the inversion symmetry, thus a

contribution of one monolayer will dominate over the other and SHG occurs. It is important to note that, the asymmetry of the hBN environment due to the different thicknesses of the top and bottom layers and 'vertical' disorder caused, e.g., by the fluctuations of the top layer thickness, can result in the SHG at $F_z = 0$ at the A-exciton, e.g., due to the difference of the intralayer A-exciton energies and their nonradiative dampings. We stress that the six-fold symmetry of the SHG signal (see supplement) clearly rules out, e.g., an in-plane asymmetry due to the strain or in-plane fields[20].

In summary, by varying excitation laser energy, dielectric environment, and applied electric fields we show strong and tunable exciton-mediated SHG in 2H $MoS_2$ bilayers that can surpass the off-resonance SHG signal in monolayers. Drastic enhancement of the bilayer SHG amplitude is observed when twice the laser energy is in resonance with the excitonic transitions. The SHG signal outside exciton resonances remains orders of magnitude smaller than the monolayer signal, indicating a very small intrinsic $\chi$ value from broken inversion symmetry. At the interlayer exciton resonance, we tune the SHG signal by over an order of magnitude in electric fields applied perpendicular to the layer and demonstrate that the spectral width of the SHG resonance increases. With our model calculations, we relate the SHG in applied electric fields to the Stark splitting of the interlayer exciton and its mixing with intralayer excitons, which results in a quadratic dependence of the SHG amplitude on the applied electric field.

Our scheme for tuning SHG based on interlayer excitons with a permanent static electric dipole can be applied to a variety of other systems with strong interlayer exciton resonances, such as homobilayer $MoSe_2$[66]. Very importantly it has been shown recently that several heterobilayer systems host interlayer excitons with static electric dipoles and high optical oscillator strength, for example, $MoSe_2/WS_2$ with hybridized conduction states[61,62], $WSe_2/WS_2$ with hybridized valence states[63], $MoS_2/WS_2$ with electrically tunable valence state hybridization[64] and similar predictions of electronic state hybridization for specific twist angles in $MoTe_2/MoSe_2$[65]. It can be expected that also in these and other systems SHG can also be efficiently tuned via the Stark effect, as demonstrated here for homobilayer $MoS_2$.

## Methods

**Sample fabrication.** The 90-nm-thick $SiO_2/Si$ substrates were cleaned for 10 min in aceton and isopropanol using an ultrasonication bath and were subsequently exposed in oxygen plasma for 60 s. Bulk 2H $MoS_2$ (2D Semiconductors) was first exfoliated on Nitto Denko tape and the exfoliated areas were attached on a poly-dimethylsiloxane (PDMS) stamp, supported by a microscope glass slide. Mono-layers, 2H bilayers, and trilayers were identified based on the optical contrast under an optical microscope prior to transfer on the $SiO_2/Si$ substrate. For the hBN-encapsulated samples, hBN flakes were first exfoliated on a Nitto Denko tape from high-quality bulk crystals while the same PDMS-assisted transfer process on $SiO_2/Si$ substrates was followed. A staircase sample of monolayers, 2H bilayers and trilayers was subsequently transferred and capped in hBN. Between every transfer step, annealing at 150 °C was applied for 60 min. Optical images of hBN encapsulated but also bare $MoS_2$ in $SiO_2$ are shown in Supplementary Information. The thickness of the bottom hBN layers was selected to optimize the visibility of the interlayer exciton in the reflectivity spectra. For the electric field device, the same process was followed including the additional transfer of few-layered graphite (FLG) flakes. The stack, starting from bottom to top consists of Si, 90 nm $SiO_2$, 130 nm hBN, FLG, few-layered (≈15 nm) hBN, 2L $MoS_2$, few-layered hBN (≈20 nm), and FLG again. The bottom and top FLG are in contact with gold (Au) electrodes to apply a potential difference and generate an electric field perpendicular to the structure. The precise sequence of the complete stack from bottom to top includes hBN/FLG/hBN/2H-$MoS_2$/hBN/FLG. A schematic representation of the device is shown in Supplementary Information.

**Experimental setup.** Optical spectroscopy is performed in a home-built micro-spectroscopy setup assembled around a closed-cycle, low vibration helium cryostat with a temperature controller ($T = 4$ K to 300 K). For SHG measurements, we use ps pulses, generated by a tunable OPO synchronously pumped by a mode-locked Ti:sapphire laser. SHG signal is collected in reflection geometry. A combination of linear polarizers and halfwave plates allows the control of excitation and detection

polarization for the polarization-resolved measurements with the setup sketched in Supplementary Information. The light is focused onto the sample at $T = 4.2$ K using a microscope objective (NA = 0.8). The position of the sample with respect to the focus can be adjusted with cryogenic nanopositioners. The reflected light from the sample is sent to a spectrometer with a 150 grooves per millimeter grating. The spectra are recorded by a liquid-nitrogen cooled charged coupled device (CCD) array. For low temperature white light reflectance measurements a white light source; a halogen lamp is used with a stabilized power, focused initially on a pin-hole that is imaged on the sample. The emitted and/or reflected light was dispersed in a spectrometer and detected by a Si-CCD camera. The excitation/detection spot diameter is 1 μm, i.e., smaller than the typical size of the homo-bilayers. We obtained differential reflectivity from reflectivity spectra as ($R_{ML} - R_{sub})/R_{sub}$, where $R_{ML}$ is the intensity reflection coefficient of the sample with the $MoS_2$ layer and $R_{sub}$ is the reflection coefficient of the hBN/$SiO_2$ stack.

## Data availability
The data that support the findings of this study are available from the corresponding authors upon request.

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

## Acknowledgements

Toulouse acknowledges funding from ANR 2D-vdW-Spin, ANR MagicValley, ANR IXTASE, ANR HiLight, ITN 4PHOTON Marie Sklodowska Curie Grant Agreement No. 721394 and the Institut Universitaire de France. Growth of hexagonal boron nitride crystals was supported by the Elemental Strategy Initiative conducted by the MEXT, Japan, Grant Number JPMXP0112101001, JSPS KAKENHI Grant Number JP20H00354 and the CREST(JPMJCR15F3), JST. M.M.G. and L.E.G. acknowledge the RFBR and CNRS joint project 20-52-16303. L.E.G. was supported by the Foundation for the Advancement of Theoretical Physics and Mathematics "BASIS".

## Author contributions

S.S. and I.P. performed the measurements, the data analysis. M.M.G, L.E.G. and I.C.G. performed the model calculations. T.T. and K.W. grew the high-quality boron nitride. C.R., D.L., A.B., and L.L. mounted the optical setup. S.S., I.P., and C.R. fabricated the van der Waals stacks. X.M. and B.U. supervised the project and suggested the experiment. All authors discussed the results and commented on the manuscript at all stages.

## Competing interests

The authors declare no competing interests.
