## [Peer Review File · Nature Communications]

Reviewers' Comments:

Reviewer #1:

Remarks to the Author:

The authors report the interlayer exciton mediated SHG in bilayer MoS₂. The results are very interesting. On the other hand, I also have lots of comments for the authors' information.

(1) The authors mention the enhancement of the SHG at the intralayer exciton resonances in bilayer when $F_z=0$.

About a small built-in electric field in the sample, are the results in fig. 3b symmetric when $F_z<0$?

About the in-equivalence of the intralayer excitons: I am not quite sure about this, as it is pretty clear that the samples prepared by the authors have high quality. Nevertheless, it is hard to believe these can enable a significant enhancement in A/B excitons, much stronger than IE (which, according to the authors, introduces symmetry breaking). Similar cases for how the degeneracy (when $F_z=0$) of the interlayer exciton states can result in enhancement of SHG. These might be a crucial key to understand the physics behind fully.

(2) The authors discuss that "We uncover the excitonic origin of the highly tunable SHG response through the linear polarization dependence of the SHG signal with respect to the armchair direction of the sample", and "We have performed measurements at additional exciton resonances that give the same polarization dependence". Can the authors give more comments (i.e., (1) polarization dependence with the exciton origin for the bilayer case; (2) how the interlayer and intralayer excitons have a similar polarization dependence)?

(3) The authors mentioned that for A-exciton resonance, SHG in monolayer is an order of magnitude weaker after BN encapsulation. What about the IE and B exciton? It seems that SHG at the IE and B exciton wavelengths is significantly (by 2-orders) reduced after BN encapsulation (if comparing fig. 1e/b). On the other hand, SHG at A:2s and X resonance is significantly increased. If so, any comments?

What is the SHG intensity difference in monolayer (and tri-layer) MoS₂ after BN encapsulation?

(4) Can the authors estimate the conversion efficiency and $\chi^{(2)}$ of mono-, bi- and tri-layer MoS₂ samples (in fig., 2 a/b/c, fig. 3/a)? The peak power intensity used in the experiments is suggested to present.

(5) Did the authors carry out the temperature-dependent results (similar in Fig. 1e, Fig. 2 a-c)?

(6) The authors discuss that "In the resonant case the bilayer SHG signal reaches amplitudes comparable to the off-resonant signal from a monolayer". Actually, if you compare the tri-layer case (shown in Fig. 2d): the tri-layer SHG signal at IE wavelength significantly increases and reaches amplitudes even comparable to the on-resonant signal (in the tri-layer). This is interesting. Can the authors give more discussion on this (which might help the authors to understand the physics behinds it)?

In addition to Fig.2d, it would be nice to present the SHG ratio at IE and A exciton wavelengths as a function of the number of layers. What does the label (bilayer/monlayer=1/3) mean?

(7) What are the peaks right to IE in trilayer MoS₂ (in Fig.2c and SI Fig.3c)?

(8) The authors discuss that "Charging effects as a possible source of symmetry breaking in bilayer SHG response are discussed for WSe₂ [44], our data demonstrate that this has negligible effect on the SHG response at the interlayer exciton energy". Can the authors show the results (in particular SHG and differential reflectivity at the IE wavelength, A/A:2S/X/B excitons) in bilayer/trilayer MoS₂ at different doping/charging levels? This might be helpful, e.g., to understand the physics behinds, to confirm the authors' justification: red-shift of A-exciton in fig. 3a, and "A weaker effect of the electric field F on the A-excitons can be readily understood taking into account the fact that the A-excitons are mixed, e.g., with the interlayer B-excitons via hole tunneling".

And why the authors did not observe the doping introduced symmetry breaking in bilayer (in principle, this is physically similar to the lift degeneracy of inter-layer)?

And the authors claim that "In contrast, for the intralayer exciton we do not record any increase of the SHG amplitude as F_z is increased". Maybe change "any increase" into "any significant increase", as fig.3a shows an increase for A exciton (~ 2 -time for the max peak amplitude, but am not sure why this is not very clear in fig. 3b).

(9) Can the authors show the results till 2.15eV in fig. 3a and fig.3c? This could be very interesting to see the responses at other peaks and B exciton. This also can be important if the authors claim that the IE activity is mainly due to the mixing with B-excitons.

(10) did the authors measure the tunable gate SHG in trilayer MoS₂ at different wavelengths (similar to fig. 3a)?

(11) If we compare fig. 3c/e, it seems the spectral width of combined IE1 & IE2 in fig.3c is much broader than the width of enhancement spectrum in fig. 3e. Can the authors reduce the wavelength tuning step?

PS: I would not suggest using "amplification" in the corresponding parts. Maybe "enhancement" is better.

(12) Not fully understand the conclusion of supplementary Section VIII A SHG in an ideal centrosymmetric bilayer: "To conclude this part, in ideal centrosymmetric bilayer the SHG is possible due to the effects related to the finite wavevector of the radiation and a finite width of the bilayer. Because of this, the effect is suppressed in pristine bilayer as compared to the non-centrosymmetric monolayer." It seems evident that the SHG in bilayer reported here is not related to "the finite wavevector of the radiation and a finite width of the bilayer". Maybe the authors can comment on this?

(13) Can the authors give more details about Equation 2. Seems ref. 37 in the text is for monolayer and no exact deduction in Supplementary document.

Reviewer #2:

Remarks to the Author:

The manuscript of Shree et al. presents resonant second harmonic (SHG) generation in few layer MoS₂ with a focus on bilayers. The authors show that, though centrosymmetric, SHG in bilayer MoS₂ is present and strongly enhanced when the exciting laser is tuned into resonance with excitonic transitions. Effects of environment and differences between intra- and interlayer excitons are discussed. Most interestingly, the intralayer exciton mediated SHG is strongly enhanced by an electric field by several orders of magnitude.

The manuscript is written very well and the authors carefully examine the various contributions to the SHG. I would like to add, though, that I cannot judge the theoretical part because I don't have enough expertise in this area. I consider their findings very interesting to a broad readership and suitable for publication in Nature Communications. Nevertheless, I have some minor questions /comments:

- Page 3, left column: „The spectral width of the SHG signal is limited by the laser pulse duration (ps).“ Does this imply that it is on the order of picoseconds or the pulse duration is 1 ps? Related to that: What is pulse energy used in the experiments?

- Page 4: Clearly, the y axes in Fig.2 a, b and c have different scales. Could you indicate by how much they are rescaled (e.g. by a different factor)? It would be interesting to see how the SHG intensities between 1L, 2L and 3L compare (e.g. the maximum SHG for the intralayer X). This is difficult to judge from panel d.

- Encapsulated bilayer: "a far more symmetric dielectric environment": how thick are the hBN layers? Are they of the same thickness on both sides?

- Page 5/6: When reading this section my question was "How is the electric field applied?" Maybe an additional reference to the supplemental Material, where the corresponding sample geometry is

mentioned, would be helpful.

- Supplemental, page 1 (right column): "The thickness of the bottom hBN layers was selected to optimize the oscillator strength of the interlayer exciton" It would be good if you could explain in more detail how a changed oscillator strength affects the SHG and how the oscillator strength of the bare 2L compares to the encapsulated structure.

Reviewer #3:

Remarks to the Author:

In this manuscript, second harmonic generation from bilayer MoS₂ is investigated with the participation of interlayer exciton. The experiments are performed by tuning the wavelength of excitation laser. Both intralayer and interlayer exciton behaviors can be observed, which can also be modulated by external electric field. The mechanism becomes clearer by comparing properties of bilayer MoS₂ with those of mono- and tri-layers. Nonlinear optical theories have been derived to interpret experimental results. In my opinion, this is a solid work, and can be published in the NC. Before that, the following questions need to be addressed: 1, More information should be provided, i.e. the measured results at 77K, as well as at room temperature, and the comparison between those in the current manuscript. 2, What is the exact twist angle of the bilayer MoS₂, 60 or 180 degree? 3, How about B exciton in Fig. 3? 4, Where do the Eq. (1) come from? 5, About the experimental details, how to tune the wavelength of excitation laser of OPO, manually or automatically? Did you compare the difference between ps and fs pulse excitation?

Below the authors provide a point by point reply to the questions raised by 3 reviewers on the manuscript

“Interlayer exciton mediated second harmonic generation in bilayer MoS₂”
by Shivangi Shree et al

Referee 1 states that: *“The authors report the interlayer exciton mediated SHG in bilayer MoS₂. The results are very interesting”.*

Referee 2: *“The manuscript is written very well and the authors carefully examine the various contributions to the SHG. I consider their findings very interesting to a broad readership and suitable for publication in Nature Communications”.*

And referee 3 wrote: *“In my opinion, this is a solid work, and can be published in the NC.”.*

Reply to reviewer 1

The authors report the interlayer exciton mediated SHG in bilayer MoS₂. The results are very interesting. On the other hand, I also have lots lots of comments for the authors’ information.

Reply: We thank the referee for the positive and detailed evaluation of our work.

(1) The authors mention the enhancement of the SHG at the intralayer exciton resonances in bilayer when $F_z=0$.

About a small built-in electric field in the sample, are the results in fig. 3b symmetric when $F_z<0$)?

About the in-equivalence of the intralayer excitons: I am not quite sure about this, as it is pretty clear that the samples prepared by the authors have high quality. Nevertheless, it is hard to believe these can enable a significant enhancement in A/B excitons, much stronger than IE (which, according to the authors, introduces symmetry breaking). Similar cases for how the degeneracy (when $F_z=0$) of the interlayer exciton states can result in enhancement of SHG. These might be a crucial key to understand the physics behind fully.

We thank the referee for raising these interesting points. Let us begin with the first part of comment (1).

We have performed laser energy dependent SHG in MoS₂ bilayers for 3 different types of samples: (i) on SiO₂; (ii) in hBN but without gate electrodes; (iii) in hBN with electrodes. For all three types of samples, we see strong enhancement of SHG at exciton resonances in the absence of any external electric field. For the sample with electrodes

the experimental plan was first to monitor the sample's response by performing reflectivity measurements (Fig. 3c) and then work in the same regime for the SHG experiments for small electric fields, $F_z < 0.2 \text{ MV/cm}$. Indeed, we performed electric-field dependent measurements in both directions (i.e. for $F_z > 0$ and $F_z < 0$). We find that both reflectivity and SHG results are not symmetric between the two directions. In particular, when we applied negative values of F_z , the required voltages to observe a Stark shift were significantly larger. It is always possible that charges can accumulate at the interfaces of the different layers of the device and screen the field asymmetrically. Another explanation can be linked to the different contact quality of the top / bottom few-layered graphene and the Au electrodes or the difference of the top and bottom hBN thickness. Consequently, we only focused on the measurements for positive F_z : the Stark splitting between the two IE components as a function of the applied voltage gives reasonable values of the IE dipole moments ($\sim 0.4 \text{ e}\cdot\text{nm}$), in very good agreement with recent reports (N. Leisgang et al., *Nature Nanotechnology* 15, 11, 901-907, 2020). The possibility of a small built-in electric field cannot be excluded, but the A-exciton, for example, is not very sensitive to the applied electric field over the range we studied, see reply to Point 8.

Regarding the second part of comment (1) and the intralayer excitons; it is certain that in the ideal case of a defect-free bilayer with a totally symmetric dielectric environment, no SHG signal is expected. This is true regardless of the excitation energy being in resonance with excitonic states or not, because the crystal is inversion symmetric in a 2H-bilayer (the radiation wavevector induced contribution to the SHG is negligible, see discussion in the manuscript). Based on the optical response of the samples studied here, we agree with the referee that they are of high quality. However, there is still a finite defect density, impurities and interface contaminants between the bilayer and from top/bottom hBN layers that we cannot neglect. Any tiny perturbation in the bilayer caused by any of these sources can generate inhomogeneous broadening in the excitonic transitions and result in energetically inequivalent states between the top and bottom layer. This applies both to intra- and interlayer states. Also, we note that the oscillator strength of the interlayer states is smaller compared to the intralayer ones (ratio $\sim 1/5$). Therefore, it is expected that the IE will exhibit weaker SHG for $F_z=0$, especially compared to A-excitons. The strong mixing between IE and B-excitons makes the interpretation for the B-exciton SHG strength more complex. By studying samples in different dielectric environments and by applying external fields, we provide detailed information on different contributions to SHG coming from either the dielectric environment or the crystal itself.

On the theory side, we have analyzed the role of the zero-field symmetry breaking in the SHG, see the Supplemental Information, Sec. VIII, subsection B2 where we included a small splitting of B-excitons, δE_B (and the same for A-excitons). The analysis shows that indeed, at $F_z=0$ the SHG is possible due to the asymmetry of the structure, Eqs. (S19) and (S20).

(2) The authors discuss that “We uncover the excitonic origin of the highly tunable SHG response through the linear polarization dependence of the SHG signal with respect to the armchair direction of the sample”, and “We have performed measurements at additional

exciton resonances that give the same polarization dependence". Can the authors give more comments (i.e., (1) polarization dependence with the exciton origin for the bilayer case; (2) how the interlayer and intralayer excitons have a similar polarization dependence)?

We agree with the referee that we need to elaborate more on these statements. The polarization dependence of the SHG follows from the symmetry of the system. The symmetry analysis is presented in the SI, Sec. VIIIA. For the 2H homobilayer both in the absence and in the presence of electric field the SHG is expected to demonstrate the six-fold symmetry (see also Eq. (1) of the main text). Note that in the case of ideal 2H homobilayer and $F_z=0$ the SHG is possible due to the radiation wavevector effects. The same dependence is expected in the case of the $z \rightarrow -z$ mirror symmetry breaking, e.g., by the asymmetric dielectric environment. Our experiments indeed demonstrate that regardless of the excitation energy being below the optical gap (lattice symmetry) or in resonance with intra/interlayer excitonic states, the SHG modulation has the same 6-fold symmetry (see Reply Figure 1 for 1L and 2L MoS₂ for resonant and non-resonant excitation conditions).

In addition, if in the monolayer plane symmetry-breaking perturbations (e.g., the in-plane strain) were present, we could expect that their manifestations will be most pronounced

at the exciton resonances, since any optical resonance strongly enhances any response (G. Wang et al., *Physical Review Letters*, 114, 9, 097403, 2015).

(3) The authors mentioned that for A-exciton resonance, SHG in monolayer is an order of magnitude weaker after BN encapsulation. What about the IE and B exciton? It seems that SHG at the IE and B exciton wavelengths is significantly (by 2-orders) reduced after BN encapsulation (if comparing fig. 1e/b). On the other hand, SHG at A:2s and X resonance is significantly increased. If so, any comments?

What is the SHG intensity difference in monolayer (and tri-layer) MoS₂ after BN encapsulation?

The referee puts forward a very interesting point. We must note that globally the SHG efficiency dropped by roughly one order of magnitude for all the transitions in every MoS₂ sample (1L, 2L, 3L) after hBN encapsulation. However, as the referee correctly noticed, some transitions (such as the transitions labelled tentatively A2s and X in bilayers) appear comparatively stronger after encapsulation. These transitions were not resolved in the non-encapsulated samples (both in the linear absorption/reflectivity and in the SHG) possibly because of inhomogeneous broadening effects as documented in the literature on linear optics. After hBN encapsulation there are two effects that need to be considered for the total SHG signal; first, the dielectric environment becomes homogeneous so any SHG due to the SiO₂ (bottom) and air (top) asymmetry will be suppressed. Second, encapsulation with hBN gives rise to wavelength-dependent cavity effects (interference), as in C. Robert et al., *Physical Review Materials*, 2, 1, 011001, 2018. The total SHG is then an interplay between these two effects before and after encapsulation that can result into wavelength-dependent variations. Following the reviewer's comment we have added this statement to the manuscript.

(4) Can the authors estimate the conversion efficiency and $\chi(2)$ of mono-,bi- and tri-layer MoS₂ samples (in fig., 2 a/b/c, fig. 3/a)? The peak power intensity used in the experiments is suggested to present.

The peak power intensity in our experiments for 5mW average power and 1ps pulse duration of 80MHz repetition rate is ~60W. We thank the referee for this remark. We now include this information in the manuscript.

We have performed additional experiments for the estimation of the conversion efficiency and $\chi(2)$ of 1L, 2L and 3L MoS₂. Since this is a demanding and complex experiment that requires coupling of several systems (Ti-Saph laser, Optical Parametric Oscillator, wavemeter, autocorrelator, spectrometer, etc), our response combines points (4) & (5) of the referee. Thus, detailed wavelength scans were performed in 1L, 2L, 3L MoS₂ after optimization of the system at T = 295 K, now excluding polarization components that can affect the efficiency of the collected signal.

In order to quantify the SHG power generated by the sample, optical losses due to the different components of the setup must be considered. The transmission/reflection coefficients of the optical components were taken from the manufacturers' spreadsheets

for the wavelengths that lie within our SHG spectra. The optical components involved in the detection path include :

- the objective lens (Attocube, NA=0.81, ~80 % transmission)
- a set of 2 glass windows that act as beam splitters (~92% transmission)
- 3x silver protected mirrors (PF-20-03-P01, Thorlabs, ~96% reflection each)
- an achromatic doublet (AC254-050-B-ML, Thorlabs, ~99% transmission)
- a shortpass 800nm filter (measured experimentally, 93% transmission in the SHG range)
- two silver-coated mirrors in the spectrometer (Princeton Instruments, Acton SpectraPRO, SP2500, 95% reflection each), and the grating efficiency (150g/mm, blazed at 500nm with 65% absolute efficiency).

Combining all these characteristics we conclude that before the signal reaches the detector, we estimate optical losses for the SHG emission to be ~65% (depending also on the wavelength, however the dispersion does not modulate strongly in the wavelength range studied here). We also considered the response of our detector (LN/100 BR Excelon, Princeton Instruments). The quantum efficiency of our detector is 90% in our SHG wavelength range, however this value is measured at 25 °C and is expected to drop at lower operating temperatures. The gain was 2e- ADU. Overall, we estimated the lower limits of the SHG power (photons x energy/second) that left the sample, the conversion efficiency ($\frac{P_{2\omega}}{P_\omega}$) and the sheet-SHG tensor element, $\chi_{sh}^{(2)}$, values.

For the estimation of the sheet-SHG tensor element, $\chi_{sh}^{(2)}$, we used the following equation (R. Woodward et al., *2D Materials*, 4, 011006, 2016 & S. Klimmer et al., *Nat. Photon.*, 2021, both now cited in the manuscript):

$$\chi_{sh}^{(2)} = \sqrt{\frac{c^3 \epsilon_0 f r^2 \pi t (1 + n_{eff})^6 P_{2\omega}}{16\sqrt{2} S \omega^2 (P_\omega)^2}}$$

Here, c is the speed of light, ϵ_0 is the vacuum permittivity, $f = 80$ MHz is the laser repetition rate, $r = 1 \mu\text{m}$ is the focused beam radius, $t = 1$ ps is the pulse duration measured with an autocorrelator at the full width at half maximum, $n_{eff} = 1.45$ is the effective refractive index of the substrate, $P_{2\omega}$ is the SHG average power (measured after estimating all the optical losses), $S = 0.94$ is a shape factor for Gaussian pulses and P_ω is the pump average power (measured before the objective lens).

We present below the estimation of the energy dependent SHG power, conversion efficiency and $\chi_{sh}^{(2)}$ for 1L, 2L and 3L MoS₂ measured at T = 295 K. Our values show a reasonable agreement with literature reports for 1L and 3L MoS₂. Please notice that there is no *intrinsic* $\chi_{sh}^{(2)}$ for 2L MoS₂ since the second order susceptibility cannot be defined for an inversion symmetric material. A centrosymmetric medium, in the absence of the external electric field and without radiation wavevector effects, requires $P(-E) = -P(E)$, thus the polarization P must be an odd function of E and all the even-order susceptibilities vanish. Also, we need to emphasize that the conversion efficiency ($\frac{P_{2\omega}}{P_\omega}$) is linearly dependent to the incident power, so its value is linked to the provided pump power (here 3mW power for 1L and 3L and 5mW for 2L).

Reply Figure 2. SHG power, conversion efficiency and sheet-second order susceptibility for 1L-MoS₂ (top row – red color) and 3L-MoS₂ (bottom row – black color). The A, B and interlayer (IE) exciton resonances are labelled

The values of the SHG power are comparable between 1L and 3L and lie on the order of few to 10fW, depending on the resonant or non-resonant pumping. The conversion efficiency also varies as a function of the energy and is on the order of 10^{-12} for 3mW pumping power. The sheet-second order susceptibility is $\sim 10^{-20}$ m²/V while it can be quickly converted to the bulk-like second order susceptibility ($\chi^{(2)}$) just by dividing $\chi_{sh}^{(2)}$ with the thickness of the material (e.g., 0.65 nm for 1L-MoS₂). In this case, $\chi^{(2)}$ is within the range of $\sim 10^{-11}$ m/V, 10^{-10} m/V depending on the pumping energy.

We now turn to the estimation of the SHG power and conversion efficiency of 2L-MoS₂ (see figure below). In agreement with our wavelength dependent experiments at T = 4 K shown in the main text, we see that the values of the SHG power and conversion efficiency are at least one order of magnitude smaller compared to 1L & 3L MoS₂. Please, note that the conversion efficiency in the 2L-MoS₂ is given for 5mW pumping power.

Reply Figure 3

We have now included the above figures together with a description of the estimation of the SHG power, conversion efficiency and the sheet-susceptibility in the supplementary material following the reviewer's question.

(5) *Did the authors carry out the temperature-dependent results (similar in Fig. 1e, Fig. 2 a-c)?*

Temperature-dependent experiments were not initially planned because thermal broadening effects are expected to act on the excitonic transitions, limiting the distinguishability of particular transitions, such as the interlayer excitons (IE) in bilayers. This is the reason our main experiments were conducted at $T = 4\text{K}$. Also, electric field dependent experiments are challenging at room temperature because of too many channels for leakage currents in our gated device, where we aim to apply static fields. Following reviewer's question we have now performed wavelength dependent SHG experiments in 1L, 2L and 3L MoS_2 at $T = 295\text{ K}$, presented in Reply Figure 4. ***Our main result is that the excitonic enhancement of the SHG signal also applies to room temperature experiments, when twice the laser energy is tuned into resonance with interlayer and intralayer excitons in 1,2 and 3L MoS_2*** , as can be clearly seen in Reply Figure 4. We thank the reviewer for motivating these new experiments. We summarize these new results in a concise paragraph in the main text and show the room temperature data from reply figure 4 in the supplement.

We find that the excitonic resonances appear broader at $T = 295\text{ K}$ than at $T=4\text{K}$. For instance, in 1L- MoS_2 the FWHM of the A-exciton is $\sim 35\text{meV}$ at $T = 295\text{ K}$, much broader compared to 6 meV measured at $T = 4\text{ K}$. In addition, the IE in 2L- MoS_2 (middle SHG spectrum in the Reply Figure 4), is now merged into the broader A2s and X states. As the resonances red-shift as a function of temperature, we notice that their relative strength is also affected possibly due to cavity effects, as shown already in temperature dependent reflectivity measurements on these type of structures, see <https://journals.aps.org/prb/abstract/10.1103/PhysRevB.99.035443>.

Reply Figure 4

(6) The authors discuss that “In the resonant case the bilayer SHG signal reaches amplitudes comparable to the off-resonant signal from a monolayer”. Actually, if you compare the tri-layer case (shown in Fig. 2d): the tri-layer SHG signal at IE wavelength significantly increases and reaches amplitudes even comparable to the on-resonant signal (in the tri-layer). This is interesting. Can the authors give more discussion on this (which might help the authors to understand the physics behinds it)?

Although twice the excitation energy of 1.005eV is very close to the IE energy of a bilayer (Figure 2d), it is also fairly close to the IE of the trilayer (please see the blue line

in Figure 2b,c). This means that also for the trilayer, this energy corresponds to a *quasi-resonant* condition. In contrast, for this energy there are no real states in monolayers as we are in energy between the A- and B-exciton. Noteworthy, the trilayer is not centrosymmetric (structures with odd number of layers do not possess an inversion centre), thus the SHG in the trilayer is intrinsically possible even in the absence of the F_z , radiation wavevector effects and symmetry breaking. Combining strong intrinsic SHG with quasi-resonant excitation for the trilayer leads to an overall strong trilayer SHG for this particular excitation energy.

In addition to Fig.2d, it would be nice to present the SHG ratio at IE and A exciton wavelengths as a function of the number of layers. What does the label (bilayer/monolayer=1/3) mean?

As IEs do not form in monolayers, comparison can be made only for bilayers and trilayers. The SHG ratio between IE and A exciton resonances (I^{SHG}_{IE}/I^{SHG}_A) for bilayers and trilayers is 0.25 and 0.5, respectively. For this ratio on the trilayer we have considered the stronger exciton $A_{L1,3}$ of the top and bottom layer (for details please see reply to comment (7) and I. Gerber et al., Physical Review B, 99, 3 035443, 2019).

The label *bilayer/monolayer=1/3* means that when $E_L = 1.005\text{eV}$ (close to the IE state of bilayers) the intensity ratio between the SHG of a bilayer (inversion symmetric) and a monolayer (inversion asymmetric) is comparable. Our point here is to demonstrate that for specific excitation energies, bilayers that possess an inversion center can exhibit a comparable SHG to the monolayers without the presence of any external fields. We have moved the ratio from the panel and mention it in the figure legend.

(7) What are the peaks right to IE in trilayer MoS₂ (in Fig.2c and SI Fig.3c)?

Trilayer MoS₂ is a very rich system in terms of formation of different excitonic complexes. Besides the splitting of the A-exciton into A_{L2} and $A_{L1,3}$ (due to the different band structure and dielectric environment experienced by the top and bottom layers compared to the middle one) additional complexes form in the energy range between the A and B excitons. Recent reports have demonstrated that interlayer excitons with large oscillator strengths and static dipole moments can also form (I. Gerber et al., Physical Review B, 99, 3 035443, 2019 & N. Leisgang et al., Nature Nanotechnology 15, 11, 901-907, 2020). In SI Fig.3c, we assign the different interlayer excitons as IE and IE*. As the referee correctly noticed, there are additional peaks with their assignment remaining unclear, so far. For instance, it is possible that the peaks at 2.03eV and 2.06eV are the 2s states of A_{L2} and $A_{L1,3}$. However, high magnetic field experiments (in fields of tens of Tesla) are required to verify this hypothesis by measuring the diamagnetic shift. More in-depth studies on trilayers are certainly needed in the future in a system with so many different excitonic transitions. Our target here was to discuss the interlayer exciton impact on SHG for the first time, so we focused on the more clear-cut knowledge about the bilayer.

(8) The authors discuss that “Charging effects as a possible source of symmetry breaking in bilayer SHG response are discussed for WSe₂ [44], our data demonstrate that this has negligible effect on the SHG response at the interlayer exciton energy”. Can the authors show the results (in particular SHG and differential reflectivity at the IE wavelength, A/A:2S/X/B excitons) in bilayer/trilayer MoS₂ at different doping/charging levels? This might be helpful, e.g., to understand the physics behinds, to confirm the authors’ justification: red-shift of A-exciton in fig. 3a, and “A weaker effect of the electric field F on the A-excitons can be readily understood taking into account the fact that the A-excitons are mixed, e.g., with the interlayer B-excitons via hole tunneling”.

And why the authors did not observe the doping introduced symmetry breaking in bilayer (in principle, this is physically similar to the lift degeneracy of inter-layer)?

We thank the referee for this detailed comment. We have reformulated the misleading sentence regarding the discussion on charging effects. We agree that a doping-induced symmetry breaking could amplify the SHG. However, it cannot result into a Stark splitting of the two degenerate interlayer excitons. For that we need a static out-of-plane electric field. In Figure 3c we demonstrate that there is a pronounced Stark shift and splitting of the two interlayer excitons for the electric field value applied in our SHG experiments. This observation strongly supports that the strong SHG enhancement at the IE energy originates from lifting the degeneracy between the two IEs and not from doping effects. To make this point clearer we provide below details on our device, see also reply figure 8 for a schematic.

Our electric-field device is not capable of tuning the charge density of the bilayer but only applies an electric field perpendicular to the plane. The few-layered graphene (FLG) top and bottom electrodes (SI part I) are used to generate a vertical potential difference and apply the electric field while the bilayer is not gated (i.e., there is no direct electrical contact on the bilayer). However, for high voltages, uncontrollable doping can occur even though the bilayer is not gated. Consequently, we selectively worked in the low voltage regime (electric fields ≤ 0.17 MV/cm) where charging effects are negligible and only neutral excitons are observed. By comparing the SHG experiments with reflectivity, we ensured that the applied electric field acts on the exciton dipole and the signature of the Stark effect on the interlayer exciton is visible in both linear and non-linear optical spectroscopy. It is also important to note that compared to previous experiments on symmetry breaking (J. Klein, et al., *Nano letters* 17, 1, 392-398 2017), the electric field we apply in our work is orders of magnitude smaller. Our experimental results demonstrate that the interlayer excitons react strongly to this small field (SHG amplitude and frequency tuning via Stark effect) whereas the intra-layer A-exciton remains practically unaffected (see also comparison of the SHG enhancement between inter- and intralayer states as a function of the electric field in point (9)).

And the authors claim that “In contrast, for the intralayer exciton we do not record any increase of the SHG amplitude as Fz is increased”. Maybe change “any increase” into “any significant increase”, as fig.3a shows an increase for A exciton (~2-time for the max peak amplitude, but am not sure why this is not very clear in fig. 3b).

The referee is correct. Indeed, there is a slight increase for A-exciton (less than 2-fold) and we have changed the sentence from “any increase” to “any significant increase” in the text. The reason it is not very clear in Figure3b is due to the scaling of the y-axis adapted to the quadratic increase measured for the IE. Below we show a y-axis rescaled version of Figure3b to demonstrate the behavior of the A-exciton clearly.

(9) Can the authors show the results till 2.15ev in fig. 3a and fig.3c? This could be very interesting to see the responses at other peaks and B exciton. This also can be important if the authors claim that the IE activity is mainly due to the mixing with B-excitons.

In Reply Figure 6b we show the results of Figure 3a from the main manuscript including the additional excitonic complexes at higher energies. As one can see, there is a 25-fold enhancement for the IE, while a smaller enhancement occurs for A2s (4-fold) and X (2.5-fold) peaks. The enhancement of the B-exciton is less than 2-fold, similar to the A-exciton. The strong enhancement of the SHG at the IE resonance for $F_z=0.17$ MV/cm is due to lifting the degeneracy of the two IE components that do not cancel anymore. Regarding the reflectivity measurements of Figure 3c, we plot below (right) the complete spectra for the three different electric field magnitudes as requested by the referee. It is apparent that the most drastic effect occurs for the IE, which splits into two components (IE_1 & IE_2), while A and B excitons remain unaffected. Details about the B-exciton and the interaction between IEs and B-excitons can be found in our recent work N. Leisgang et al., Nature Nanotechnology 15, 11, 901-907, 2020.

(10) did the authors measure the tunable gate SHG in trilayer MoS₂ at different wavelengths (similar to fig. 3a)?

We did not perform electric-field dependent SHG measurements in the trilayers because of the complexity of this system hosting several excitonic transitions in a narrow energy range. We focused on the bilayer system due to its inversion symmetric structure and the clear interlayer exciton signature lying between A and B excitons. However, this is a very interesting point for future experiments with emphasis on trilayers.

(11) If we compare fig. 3c/e, it seems the spectral width of combined IE1 & IE2 in fig.3c is much broader than the width of enhancement spectrum in fig. 3e. Can the authors reduce the wavelength tuning step?

PS: I would not suggest using “amplification” in the corresponding parts. Maybe “enhancement” is better.

We thank the reviewer for the suggestion to replace the word “amplification” with “enhancement”, which is indeed more appropriate. This change has now been implemented in the manuscript. The globally broader spectral width of the transitions in reflectivity (Fig.3c) as compared to the SHG envelope (Fig.3e) could be linked to the larger spotsize of the white light source in reflectivity, which can result in larger inhomogeneous broadening. Our message in Fig.3e is to show that the SHG response gets gradually

spectrally broader as we go from $0 \rightarrow 0.07 \rightarrow 0.13$ MV/cm. We find in SHG the same trend as in the reflectivity data in Fig.3c., which we thought is instructive for the reader.

(12) Not fully understand the conclusion of supplementary Section VIII A SHG in an ideal centrosymmetric bilayer: “To conclude this part, in ideal centrosymmetric bilayer the SHG is possible due to the effects related to the finite wavevector of the radiation and a finite width of the bilayer. Because of this, the effect is suppressed in pristine bilayer as compared to the non-centrosymmetric monolayer.” It seems evident that the SHG in bilayer reported here is not related to “the finite wavevector of the radiation and a finite width of the bilayer”. Maybe the authors can comment on this?

We fully agree with the reviewer that for the experimentally studied samples the symmetry breaking effects are needed to describe the SHG at $F_z=0$, while radiation wavevector effects are minor. The goal of our theory section in the SI is to discuss all possible mechanisms of SHG which can be also present in higher quality and even more symmetric bilayer samples, with the aim to lay foundation for describing further experiments and also reaching the broad readership of the Nature Communications. We have added clarifying phrases in the end of the Sec. VIIIA in the Supplement.

(13) Can the authors give more details about Equation 2. Seems ref. 37 in the text is for monolayer and no exact deduction in Supplementary document.

Following the referee’s comment, we have extended the derivation of Eq. (2) in the Supplement, see new Eq. (S13) and discussion around it for the benefit of the reader.

Reply to reviewer 2

The manuscript of Shree et al. presents resonant second harmonic (SHG) generation in few layer MoS₂ with a focus on bilayers. The authors show that, though centrosymmetric, SHG in bilayer MoS₂ is present and strongly enhanced when the exciting laser is tuned into resonance with excitonic transitions. Effects of environment and differences between intra- and interlayer excitons are discussed. Most interestingly, the intralayer exciton mediated SHG is strongly enhanced by an electric field by several orders of magnitude.

The manuscript is written very well and the authors carefully examine the various contributions to the SHG. I would like to add, though, that I cannot judge the theoretical part because I don’t have enough expertise in this area. I consider their findings very interesting to a broad readership and suitable for publication in Nature Communications.

We thank the referee for the thorough examination of our manuscript and the encouraging comments. We are delighted that the referee finds our main finding on interlayer mediated SHG most interesting.

Nevertheless, I have some minor questions /comments:

- Page 3, left column: „The spectral width of the SHG signal is limited by the laser pulse duration (ps).” Does this imply that it is on the order of picoseconds or the pulse duration is 1 ps? Related to that: What is pulse energy used in the experiments?

The referee is correct, we need to clarify this. The pulse duration is 1 ps. We have now corrected this in the manuscript. The pulse energy used in the experiments is on the order of ~0.1nJ (repetition rate of the laser is 80 MHz).

- Page 4: Clearly, the y axes in Fig.2 a, b and c have different scales. Could you indicate by how much they are rescaled (e.g. by a different factor)? It would be interesting to see how the SHG intensities between 1L, 2L and 3L compare (e.g. the maximum SHG for the intralayer X). This is difficult to judge from panel d.

This is a very interesting comment, and we have improved Fig. 2 following the question of the reviewer, thank you. It is important to demonstrate a comparable scaling between the three cases. For clarity, we present in Reply Figure 7a a comparison between the A-exciton resonances of the three cases studied, under the same experimental conditions. While the inversion asymmetric layers (1L & 3L) have the same order of magnitude SHG strength, the inversion symmetric bilayers exhibit roughly one order of magnitude weaker SHG. As shown in Reply Figure 7b we have updated Figure 2 a,b,c in the manuscript with the correct scaling in the y-axis.

Plotting figure 2 with comparable scales brought to light an interesting detail for comparing SHG at the A-exciton resonance for monolayers with trilayers: The A exciton resonance in trilayers for the middle layer L2 is redshifted compared to the A exciton resonance for the outer layers L1 and L3. Because of this shift, these contributions do not cancel each other as could be expected from symmetry, but we can measure them separately. The A exciton of the monolayer gives 1500 counts (Fig 2a) , the A exciton of L2 gives also 1500 counts and at the energy of A for L1 and L3 we get 3000 counts (Fig 2c). If for the trilayer the A-excitons in all 3 layers would be degenerate, we would expect to measure only roughly 3000-1500=1500 counts. But in an MoS₂ trilayer this degeneracy among the A-excitons is lifted and we record in total 4500 counts. This finding is added as a comment to the main text.

- Encapsulated bilayer: “a far more symmetric dielectric environment”: how thick are the hBN layers? Are they of the same thickness on both sides?

Thank you for raising this interesting point. The top and bottom hBN layers do not have the same thickness in the encapsulated sample without the contacts. The bottom hBN is $\sim 130\text{nm}$ while the top is $\sim 15\text{nm}$. From the point of view of the exciton, compared to the Bohr radius (order of 1-2nm), the dielectric environment experienced by the excitons will be far more symmetric after encapsulation. From the point of view of the complete crystal stacking, it is possible that the same hBN thickness on both sides could result into even weaker SHG of the bilayer. The reason we have used uneven top and bottom hBN thickness is to optimize the visibility of the bilayer and maximize the amplitude of the excitonic transitions in reflectivity experiments – crucial for this study to assign the various transitions, see methods described in C. Robert et al., *Physical Review Materials*, 2, 1, 011001, 2018.

- Page 5/6: When reading this section my question was “How is the electric field applied?” Maybe an additional reference to the supplemental Material, where the corresponding sample geometry is mentioned, would be helpful.

Indeed, we need to elaborate more on the device structure and how the electric field was applied. We have now added the schematic representation of the device shown below, in the Supplementary material and modified the description of the device.

The stack -starting from bottom to top- consists of Si, 90nm SiO₂, 130nm hBN, few-layered graphite (FLG), few-layered (~15nm) hBN, 2L MoS₂, few-layered hBN (~20nm) and FLG again. The bottom and top FLG are in contact with gold (Au) electrodes to apply a potential difference and generate an \vec{E} field perpendicular to the structure.

- Supplemental, page 1 (right column): “The thickness of the bottom hBN layers was selected to optimize the oscillator strength of the interlayer exciton” It would be good if you could explain in more detail how a changed oscillator strength affects the SHG and how the oscillator strength of the bare 2L compares to the encapsulated structure.

We thank the referee for putting forward this comment. In an encapsulated structure, cavity effects (i.e. interference) are determined by the top and bottom hBN thickness. The visibility of the excitonic resonances depends critically on the hBN and SiO₂ layer thickness in the sample due to thin film interferences (C. Robert et al., *Physical Review Materials*, 2, 1, 011001, 2018; H.H. Fang, et al., *Phys. Rev. Lett.*, 123, 067401, 2019). Our target was to optimize the visibility of weaker exciton states (in terms of oscillator strength) by the selection of the top and bottom hBN thickness for reflectivity experiments. This would assist with the assignment of the excitonic states in the wavelength dependent SHG experiments by comparing them with reflectivity spectra (i.e. Figure 1 e,f). To make this point clear and remove the confusion on the oscillator strength we have rewritten the sentence in the supplementary material, page 1 right column, to “The thickness of the bottom hBN layers was selected to optimize the visibility of the interlayer exciton in the reflectivity spectra.”

Reply to reviewer 3

In this manuscript, second harmonic generation from bilayer MoS₂ is investigated with the participation of interlayer exciton. The experiments are performed by tuning the wavelength

of excitation laser. Both intralayer and interlayer exciton behaviors can be observed, which can also be modulated by external electric field. The mechanism becomes clearer by comparing properties of bilayer MoS₂ with those of mono- and tri-layers. Nonlinear optical theories have been derived to interpret experimental results. In my opinion, this is a solid work, and can be published in the NC.

We appreciate the evaluation of our work by the referee and the support for publication.

Before that, the following questions need to be addressed:

1, More information should be provided, i.e. the measured results at 77K, as well as at room temperature, and the comparison between those in the current manuscript.

We thank the referee for this suggestion. We have taken this comment seriously, questions about room temperature were also raised by reviewer 1, we performed additional experiments accordingly. Below, we reproduce here our reply to point (5) of reviewer 1. :

Temperature-dependent experiments were not initially planned because thermal broadening effects are expected to act on the excitonic transitions, limiting the distinguishability of particular transitions, such as the interlayer excitons (IE) in bilayers. This is the reason our main experiments were conducted at $T = 4\text{K}$. Also, electric field dependent experiments are challenging at room temperature because of too many channels for leakage currents in our gated device, where we aim to apply static fields.

We have now performed wavelength dependent SHG experiments in 1L, 2L and 3L MoS₂ at $T = 295\text{ K}$, presented in Reply Figure 4. ***Our main result is that the excitonic enhancement of the SHG signal also applies to room temperature experiments, when twice the laser energy is tuned into resonance with interlayer and intralayer excitons in 1,2 and 3L MoS₂***, as can be clearly seen in Reply Figure 4. We thank the reviewer for motivating these new experiments. We summarize these new results in a concise paragraph in the main text and show the room temperature data in the supplement.

We find that the excitonic resonances appear broader at $T = 295\text{ K}$ than at $T=4\text{K}$. For instance, in 1L-MoS₂ the FWHM of the A-exciton is $\sim 35\text{meV}$ at $T = 295\text{ K}$, much broader compared to 6 meV measured at $T = 4\text{ K}$. In addition, the IE in 2L-MoS₂ (middle SHG spectrum in the Reply Figure 4), is now merged into the broader A_{2s} and X states. As the resonances red-shift as a function of temperature, we notice that their relative strength is also affected possibly due to cavity effects, as shown already in temperature dependent reflectivity measurements on these types of structures, see

<https://journals.aps.org/prb/abstract/10.1103/PhysRevB.99.035443>.

Reply Figure 4 – copy

2, What is the exact twist angle of the bilayer MoS₂, 60 or 180 degree?

The MoS₂ bilayers studied here possess 2H stacking order, as can be concluded, for example, from the energy separation we measure between the A and B excitons, as in L. Paradisanos et al, Nature Comms 2020. The bilayer has been exfoliated from 2H bulk and has not been assembled manually. 2H stacking corresponds to a twist angle of 60°. Since the stacking order modulates periodically for twist angled of 60°, it means that twist angles of both 60° and 180° are indistinguishable and belong to the 2H stacking, in contrast to 0° and 120° where the stacking order is 3R (with broken inversion symmetry).

3, How about B exciton in Fig. 3?

We thank the referee for this comment. In reply figure 9 we show the results of Figure 3a including the additional excitonic complexes at higher energies. As one can see, there is a 25-fold enhancement for the IE, while a milder enhancement occurs for A_{2s} (4-fold) and X (2.5-fold) peaks. The enhancement of the B-exciton is less than 2-fold, similar to the A-exciton.

4, Where do the Eq. (1) come from?

Equation (1) is derived from the symmetry arguments: we have determined the quadratic combinations of the field components ($E_x^2 - E_y^2, 2E_xE_y$) which transform as the vector components (P_x, P_y). Following the reviewer's question, we have added a brief comment about it in the main text after Eq. (1) and also provided a reference with more detailed symmetry analysis.

5, About the experimental details, how to tune the wavelength of excitation laser of OPO, manually or automatically? Did you compare the difference between ps and fs pulse excitation?

The wavelength of the excitation laser of the OPO was tuned manually – so each data point we present is a separate experiment. We haven't compared any differences between ps and fs pulse excitation. The pulse duration used here was 1ps and was always measured using a part of the beam aligned to an auto-correlator.

For sample systems with different exciton transitions spectrally close in energy, ps pulses allow better spectral selectivity. fs pulses might overlap spectrally with more than one exciton resonance and interpretation of the spectral dependence more challenging.

Reviewers' Comments:

Reviewer #1:

Remarks to the Author:

i carefully check the reply letter and the revised manuscript. it appears that the authors have responded/addressed to most of the key issues of all questions from the reviewers. For your decision, I am happy to recommend the publication.

Reviewer #2:

Remarks to the Author:

The authors have addressed all my questions and comments most satisfactorily. I am happy to recommend publication in Nature Communications.

Reviewer #3:

Remarks to the Author:

The manuscript was improved a lot. It could be considered for publication in Nature Communication.